# The ^31^P Spectral Modulus (PSM) as an Assay of Metabolic Status

**DOI:** 10.3390/biology14020152

**Published:** 2025-02-02

**Authors:** Jack V. Greiner, Tamara I. Snogren, Thomas Glonek

**Affiliations:** 1Department of Ophthalmology, Harvard Medical School, Boston, MA 02114, USA; 2Schepens Eye Research Institute of Massachusetts Eye & Ear, Boston, MA 02114, USA; 3Clinical Eye Research of Boston, Winchester, MA 01890, USA; snogrentamara@gmail.com (T.I.S.); tglonek@rcn.com (T.G.)

**Keywords:** adenosine triphosphate, ATP, high-energy phosphates, low-energy phosphates, metabolic health, phosphate metabolism, NMR, nuclear magnetic resonance, phosphorus-31 nuclear magnetic resonance, ^31^P spectral modulus

## Abstract

Energy metabolism can be measured utilizing phosphorus-31 nuclear magnetic resonance (^31^P NMR) spectroscopy in both intact living and extracts from cells, tissues, and organs. The ^31^P NMR spectrum, comprised of the organophosphates involved in intermediary metabolism, allows for the calculation of a ^31^P spectral modulus (PSM). As such, the PSM measurement permits an assay of overall metabolic health and disease status and can provide assessments of declining tissue health as well as improvements in disease processes. This numerical PSM is the ratio of the high-energy phosphate to the low-energy phosphate spectral bands. The PSMs of normal cases, generally >1.6, are significantly greater than either stressed or diseased cases at <1.3. This study presents the foundations and fundamentals of the PSM, a living index of tissue health. The PSM can be employed for the measurement of previously published ^31^P NMR spectra as well as interim laboratory copies of spectra. Since the PSM only requires a measurable integral curve, it can be generated from minimally detectable sample concentrations of metabolites whether determined in vitro or in vivo. Thus, the PSM can allow the in vivo assaying of metabolic status using already developed ^31^P NMR surface-coil technology.

## 1. Introduction

Phosphorus nuclear magnetic resonance (^31^P NMR) spectral profiles of a large variety of normal living tissues, cells, and organs are remarkably similar in appearance, both qualitatively and quantitatively [1,2,3,4]. Moreover, such electromagnetic profiles, as customarily displayed in the laboratory, are seen to consist of two prominent spectral bands, a lower (downfield) chemical shift (δ scale [5]) band and a higher (upfield) chemical shift band separated by a segment of approximately 0.2 δ. Within this segment is an inflection point in the spectral integral curve at ca. −0.13 δ [4] (Figure 1). This segment exhibits trace or no phosphorus signals, thus conveniently dividing the spectral profile into two distinct spectral bands. The spectral bands arise from the chemistry of the constituent phosphate chemical groups, the low-energy phosphate band consisting of orthophosphate ester signals and the high-energy phosphate band consisting of resonances from phosphate anhydrides, e.g., adenosine triphosphate (ATP), and related anhydride-like molecules, such as creatine phosphate and arginine phosphate [2]. In any given spectrum, these bands do not overlap.

Metabolic analysis of these phosphate bands and their components is facilitated through the use of the spectral integral, which determines spectral signal areas and which is an analytical function of the NMR spectrometer. The spectral integral quantifies the spectral bands and their constituent components, and (quantitative) metabolic indices, such as the phosphorus spectral modulus (PSM), which is the simple ratio: (high-energy phosphate signal area)/(low-energy phosphate signal area).

This metabolic index has been used for specific biological preparations [4]. In this study, the PSM is applied to a wide variety of specimens (*n* = 347) to demonstrate the robustness of the index in determining specimen high-energy metabolic status. The index can be applied as a ^31^P spectroscopy metabolic adjunct to NMR resonance imaging (^1^H or proton MRI) for the determination of organ and tissue high-energy phosphate metabolism as a measure of health.

The PSM is calculated using the integral curve or area under the signals comprising the entire spectrum determined by ^31^P NMR spectroscopy. The organophosphates so determined are involved in intermediary metabolism including the principle high-energy metabolite, ATP. The metabolic status is a determiner of the health of cells, tissues, and organs and thus can be used to monitor therapeutic and non-therapeutic health via the integral curve. The quantitative signal areas obtained represents information at the cellular level arising from the gross composition of cellular membrane intermediates, phosphorylated intermediates of intermediary metabolism, and intermediary polynucleotide biochemistry, as well as the vitamin nucleotide enzymatic co-factors of metabolic processes.

### 1.1. ^31^P Spectral Metabolites

The phosphatic metabolites included within the spectral range incorporated into the PSM computation (from 10 to −25 on the δ scale) must contain a phosphate chemical functional group, either as an orthophosphate ester or as an esterified condensed (anhydride) phosphate, such as ADP and ATP chain phosphates (usually esterified on one end). The exceptions are molecules, such as phosphocreatine and functionally related molecules, e.g., phosphoarginine, which have chemical properties similar to those of the phosphate anhydrides. The ^31^P spectrum has very few biochemical exceptions, e.g., the phosphonates, principally derived from marine sources, and phosphine and other reduced forms as products of microbial activity under anaerobic conditions [6]. Such molecules have ^31^P chemical shifts well outside the range used in the PSM computation. As many as 43 organophosphates of intermediary metabolism have been detected and measured spectroscopically in a single assay using ^31^P NMR (Table 1) [3].

### 1.2. Phosphorus-31 (^31^P) NMR Spectral Signals

The ^31^P NMR spectrum is comprised of signals derived from organophosphate metabolites that resonate in the laboratory spectrometer’s electromagnetic field from 10 to −25 on the δ scale (Figure 1). The precise resonance shift position of each signal in the ^31^P NMR spectrum is a characteristic physiochemical marker of the metabolites present. The chemical identification of the resonance signals is based on multiple physicochemical criteria. Such criteria include the standardized chemical shift position of the peak signal, peak super positioning with an added compound of known identity, the demonstration of co-migration with a known compound, the proton-coupled peak characteristics, the J coupling values of the peak multiplets, and pH-dependent resonance-shift-curve characteristics.

The organophosphates detected in the ^31^P NMR spectrum can be divided into high-energy phosphate and low-energy phosphate bands (Figure 1). The ratio of the concentration of high-energy phosphates to low-energy phosphates has been used to monitor health [7] and disease states in cell, tissue, and organ extracts [8,9,10], including tissues preserved for transplantation [11]. The principle high-energy phosphate in healthy cells, tissues, and organs is ATP, present in high (>2.3 mM) intracellular concentrations [2,12,13]. The use of high-energy phosphates, such as ATP, for the purpose of driving metabolism, results in the accumulation of low-energy phosphates. Ordinarily, in healthy tissues, the low-energy phosphates are recycled through phosphorylation reactions that directly or indirectly use ATP as the ultimate high-energy phosphate source molecule. Although ATP is most commonly known principally for its function as the essential carrier of high-energy phosphate bonds in cellular metabolism, ATP also has a major role in preventing protein aggregation [8,14]. These functions maintain both the structural and enzymatic activity of proteins. Both of these functions suggest that the measurement or assay of ATP and the other high-energy phosphates can be used to monitor health and disease states in the body. These roles permit the concentration of ATP to be a measure of, and to be used in the production of, the high-energy phosphates essential for the maintenance of cellular, tissue, and organ metabolism, including nucleic acid metabolism.

### 1.3. Adenosine Triphosphate (ATP)

Extraordinarily high concentrations of ATP have been observed in many tissues and even in unicellular organisms [7]. Since in nature, the production of molecular species, such as ATP, is conservative, it becomes obvious that high concentrations of ATP must be necessary to maintain adequate cellular, tissue, or organismal metabolism, while preventing protein aggregation [8]. Deficiency of ATP most likely would result in dysfunction. This would result from lowered ATP-dependent metabolic rates and other enzymatic dysfunctions resulting from protein insolubility [7,14]. In that the high-energy phosphate bonds within ATP are purported to be the currency of intracellular metabolism, the more recent function of ATP in metabolism at the molecular level is hypothesized to be a hydrotropic property that functions as a modulator of protein aggregation [8,14,15,16].

Since ATP normally exhibits the greatest concentration among the organophosphates intracellularly, the ratio of the high-energy phosphates to the low-energy phosphates, defined as the ^31^P spectral modulus (PSM, aka the ^31^P energy modulus) [17], can be employed as an assay to determine not only the overall metabolic health status of a tissue, but potentially also can indirectly be a measure of protein dysfunction. Additionally, the PSM indirectly can be used to determine the effects of nutritional neglect e.g., hypoglycemia [13], as well as the effects of disease [9,10], therapeutic treatment regimens, treatment modification(s), and the monitoring of function over a time-course [13,18,19].

### 1.4. The Phosphodiesters

Considering the entire spectrum of ^31^P NMR organophosphates, not all are related to energy metabolism. For example, organophosphates, such as the phosphodiesters glycerylphosphorylcholine (GPC) and glycerylphosphorylethanolamine (GPE), are related to phospholipid metabolism and usually are present in low concentrations, in contrast to the concentrations of the high-energy phosphates and related molecules, e.g., adenosine tri- and diphosphates and the dinucleotides, e.g., nicotinamide adenine dinucleotide (NAD), flavin adenine dinucleotide (FAD), etc. As such, the influence of these non-energy related molecules is more or less trivial in the grand scheme regarding the PSM [3].

### 1.5. ^31^P Spectral Modulus (PSM) Computation

The PSM can be calculated in two ways from ^31^P NMR data. The data required are either the report of measurements of all the constituents of the ^31^P NMR spectrum in mole percent phosphorus, wet-chemical quantitative values, or the overall ^31^P spectral integral curves [4]. Studies rarely report data on all of the constituents of the ^31^P NMR spectrum in healthy, stressed, or diseased cells, tissues, and organs or the ^31^P NMR spectral integral curve, and, as such, it is not possible or extremely laborious to calculate the PSM from such studies. This is because, conventionally, most investigations using ^31^P NMR technology may only report on individual organophosphates or groups of organophosphates and not the ^31^P NMR tissue spectrum as a whole—For example, ATP and other nucleotides, sugar phosphates [19], GPC in hypothermic studies [11], and the dinucleotides [20], none of which by themselves permits the calculation of the PSM. 

In contrast to our previous study, which was restricted to observations in our laboratories where sampling, preparations, and instrumental methods were rigorously controlled [17], this study calculates the PSM from ^31^P NMR spectra obtained from the literature and describes the employment of the PSM and its implications as a measure of cellular, tissue, and organ health status.

It is possible to employ the ^31^P spectral integral curve in order to calculate the PSM. Integral curves can be derived from already available and published ^31^P NMR spectra that include the range from 10 δ to −25 δ, or even in cases where such NMR spectra only contain a limited number of organophosphates reporting data segments of the high-energy phosphates and low-energy phosphates. The present study is designed to determine whether the spectroscopic findings will reveal a difference among healthy (normal) and stressed or diseased cells, tissues, and organs.

## 2. Methods

### 2.1. The ^31^P NMR Spectrum

The phosphates detected by ^31^P NMR spectroscopy, as they are ordinarily obtained using commercially available nuclear magnetic resonance spectrometers, are fortuitously plotted such that the high-energy phosphates are found in a resonance band that is discrete from a similar band containing the low-energy phosphates. These two bands do not overlap, and there is no mixing of resonances between the two bands because of a sufficiently large chemical shift gap (0.2 δ) between these two sets of functional group resonances. This convenient separation of resonance bands allows for the calculation of the PSM by simply dividing the integrated signal area assigned to the high-energy phosphates by the integral of signals assigned to the low-energy phosphates (Figure 1). The chemical shifts of these phosphatic signals in the ^31^P NMR spectrum have been tabulated (Table 1) [3].

### 2.2. Computation of the PSM (Quantification of ^31^P Spectral Metabolites)

When analysis of ^31^P NMR spectra involves only computation of the PSM, it is not necessary to employ ^1^H decoupling. ^31^P-^1^H couplings in biological phosphates are small, on the order of 5 Hz, and even with proton coupling present, there is more than enough separation between the high-energy phosphate and low-energy phosphate bands. This separation allows for clean integration of these two spectral signal groupings, permitting calculation of the PSM. The PSM serves as an assay of metabolic health with the healthy tissues having a high PSM and the diseased or stressed tissues having typically a lower or declining PSM.

^31^P spectral modulus measurements calculated from acquired ^31^P NMR spectra generate higher signal-to-noise ratios. This is because all of the detected spectral resonances are used to compute this PSM index, not just the single resonance of one of the spectral components. This provides a signal having a much greater signal-to-noise ratio that facilitates the acquisition of quality data from even the intact living tissues, with an overall limited ^31^P concentration. The PSM assay can be applied to tissue extracts as well, since the relationship between the high-energy and low-energy phosphate bands holds for all spectra derived from living tissues through perchloric acid extraction. For example, for the rabbit lens, the computed PSM from a single lens perchloric acid extract was 2.2; for the intact *ex vivo* lens obtained from the same animal, it was 2.6 [3]. As such, the PSM has application to both tissue extracts (Figure 2) [3,13], or the dynamic analysis of intact tissue over a time-course (Figure 3) [13]. Thus, the use of the PSM measurement permits an assay of overall health and disease status and further assessment of declining tissue health, or improvements in disease processes whether treated or spontaneous.

### 2.3. Tabulated ^31^P NMR Data

A literature search of published studies in Medline via PubMed was conducted using the concepts “^31^P nuclear magnetic resonance spectrum”, “cell”, “tissue”, and “organ”. No language or date limitations were used. The final search was run in December 2025. Retrieved articles that focused on the ^31^P NMR spectra of individual molecules were excluded, whereas measurements using the PSM were performed on articles presenting complete spectra. In order to reduce bias, reports of authors Greiner or Glonek were deleted from the database roster, leaving essentially cases originating from other authors. A total of 513 articles were reviewed. Retrieved articles were examined for the ^31^P NMR profiles in normal, diseased, or stressed cells, tissues, and organs under study. After title/abstract and full-text screening, 98 articles remained for inclusion in this study. This database was designed to be representative of work to December 2025. We restricted this study to published reports of ^31^P NMR spectral data, which included both quantitative and spectral data from studies from outside our laboratories that included full intact tissue or tissue extract ^31^P spectra.

In studies where digital analyses were presented in tabular form, the moduli were calculated from the tabulated numerical data. In spectra where the signal-to-noise ratio was of sufficient quality such that a reasonable baseline could be drawn (ordinarily along the abscissa or x-coordinate of a spectral graph), integrals were obtained, and the PSM computed. Integrals were calculated using the UN-SCAN-IT Graph Digitizer Software (Silk Scientific, Inc., Provo, UT, USA). This integration required that the spectral baseline be set to zero at all points within the in vivo and ex vivo spectral range. This was accomplished by selecting extreme high-field and low-field points in the spectral range (10 δ to −25 δ) that lay within the spectral noise. These points designate a spectral baseline segment that is then computed to zero to normalize the spectral curve prior to spectral integration. Spectral baseline normalization was necessary for most cases, since published spectral baselines frequently presented spectra having tilted baselines. In all the spectra that were included in the analyses (Appendix A), a minimum of approximately 5% of the total spectral length was required at the end of the low-energy and high-energy spectral bands in order to permit determination of a properly adjusted baseline prior to computation.

#### Comparing NMR Spectrometer and UN-SCAN-IT Measurements

To compare the actual measurement of the integral curve which quantifies the signal area of the low-energy and high-energy phosphate bands of the ^31^P NMR spectrum, a spectrum of the crystalline lens [13] was compared with that of the corresponding phosphate bands measured with the UN-SCAN-IT graph digitizer software. The PSM value obtained by the numeric integral derived from the NMR spectrometer of the lens spectrum was 2.1; the corresponding PSM value from the UN-SCAN-IT graph digitizer was found to be 2.4. This comparison demonstrates a close correspondence between the actual measurement of the ^31^P spectral modulus using the NMR’s computer integrals and the UN-SCAN-IT integrals.

### 2.4. Normalizing and Adjusting the NMR Spectral Baseline

In order to utilize published or acquired NMR spectra in calculation of the PSM, it is necessary to first normalize the spectral baseline. This is necessary unless the baseline of the spectrum is normalized prior to acquisition of spectral signal measurement while gathering spectral data. The spectral baseline is adjusted to the horizontal in order to yield a true zero-baseline.

We used the UN-SCAN-IT tilt correction feature in order to correct for tilted spectra. This used the Xmin and Xmax icon’s location. The x and y axis values obtained from UN-SCAN-IT were transferred to an EXCEL program where a scatter plot graph was created from those values. In order to find the baseline, an average was found utilizing the first and last *y*-axis points of the spectra. The average was then subtracted from all the *y*-axis values from each spectral signal to provide a corrected *y*-axis. Ultimately, these *y*-axis points were used to calculate the ^31^P spectral modulus.

The area of flattening between the low- and high-energy phosphate bands was first determined visually using the EXCEL scatter plot graph function. The center point of the area of flattening was established as the division of high- and low-energy phosphates. This point in the total spectrum is located between the phosphocreatine signal and the phosphodiester signal band in the region of −0.13 δ. This partitioning of the total biological phosphate band separates the low-energy and high-energy phosphate spectral values. We obtained the sum of the corrected low- and high-energy *y*-axis values for further processing. For all other computations, the *x*-axis was ignored. This is because the UN-SCAN-IT program reads spectral signal intensities down to the chemical shift scale beneath the NMR spectrum. The signal intensity is thus augmented by the amplitude between the straightened (adjusted horizontal) baseline and the line segment designating the chemical shift scale, which is a horizontal line in all of the studies examined herein. The intensity difference between these two horizontal line segments is removed by subtracting the intensity resulting from the difference between the normalized spectral baseline and the chemical shift scale line segment from all signal intensity measurements.

In this analytical procedure, the pixel intensities provided by the UN-SCAN-IT Graph Digitizer were placed in a column on an EXCEL spreadsheet. The appropriate excess intensity, which is the same value for each digitizer-generated value, was subtracted from the total digitizer value of each linear point of the normalized spectral intensity data to generate a corrected signal intensity data column. The signal intensity data column for summation by EXCEL is the corrected signal intensity for computation of the PSM [Columns E (high-intensity) and F (low-intensity)] in Appendix A.

Where the spectral quality did not permit drawing a suitable baseline, the spectral data were excluded from the modulus analysis in the present study.

### 2.5. ^31^P Spectral Figure Data

Of the 513 studies reviewed, 98 contained at least one figure and/or tabulated ^31^P NMR data suitable for spectral integration and statistical analysis. These cases (347 after removal of outliers) were entered into comprehensive Appendix A, which contains columns for differentiating normal, disease, and various conditions or stressor categories. Data sets containing mean PSM values with standard deviations and ranges are presented in Table 2; t-Test comparisons among all groups were computed are and also tabulated in Table 2.

### 2.6. Appendix A

Appendix A was constructed with the following columns: Column A, Case Number; B, Organisms/Species (a primary designation); C, Organ/Tissue/Cell (a secondary designation as defined by the author[s]); D, Nature of Gross Sample, 1 = cells, 2 = tissue, 3 = organ; E, Tissue Physiology, 1 = In vivo; 2 = Ex vivo; 3 = Extract); F, Physiological State, 1 = normal (nl); 2= stressed (stress); 3 = diseased (dx); G, High-Energy Phosphate Relative Amplitude; H, Low-Energy Phosphate Relative Amplitude; I, Spectral Modulus (PSM), HEP to LEP Ratio; J, Figure/Table Number (in the Referenced Text); K, Reference Number [21,22,23,24,25,26,27,28,29,30,31,32,33,34,35,36,37,38,39,40,41,42,43,44,45,46,47,48,49,50,51,52,53,54,55,56,57,58,59,60,61,62,63,64,65,66,67,68,69,70,71,72,73,74,75,76,77,78,79,80,81,82,83,84,85,86,87,88,89,90,91,92,93,94,95,96,97,98,99,100,101,102,103,104,105,106,107,108,109,110,111,112,113,114,115,116,117,118,119].

### 2.7. Outliers and Sorting

Before sorting the 347 cases, we removed the outliers. With respect to organism, a statistical evaluation was run for the purpose of determining outliers in the PSM data. These were as follows: for Column F = 1, outliers were any value > 5, for F = 2 & 3, outliers were any value > 4; by default, the minimum value was zero.

The method for gathering and accumulating spectral data for this study is not obvious. A systematic literature search review along with strict selection criteria was accomplished without prejudice with the exception that the quality criteria of the spectra had to be coupled with the limits of the device used to generate the spectral profiles.

## 3. Results

### 3.1. Database

Examining Column F data (Physiological State in Table 2), there are 165 normal cases, 132 stressed cases, and 50 diseased cases in the Appendix A database suitable for statistical analysis, for a total number of cases equal to 347. [Before statistical computation; there were 16 outliers for F = 1 (4.2%), 13 for F = 2 (3.4%), 3 for F = 3 (0.8%); these were removed from the active database.] After removing outliers, Organisms/Species (column B) consisted of 19 dog, 5 guinea pig, 55 human, 40 microorganisms, 86 mouse, 18 pig, 16 rabbit, 102 rat, and a variety of other species having lower tabulated counts; Organ/Tissue/Cell (column C); Nature of Gross Sample (column D) 1 (cells) 135, 2 (tissue) 144, 3 (organ) 68; Tissue Physiology (column E) 1 (in vivo) 273, 2 (ex vivo) 52, 3 (extract) 22; Physiological State (column F) 1 (normal) 165, 2 (stressed) 132, 3 diseased) 50.

### 3.2. Physiological States

After having sorted the entire Appendix A by column F (Physiological State), the statistical *t*-test was used to determine significant differences among the three designated Physiological States (Table 2).

The PSM of normal cases were significantly greater than either stressed or diseased cases or stressed and diseased cases combined. The PSM of stressed cases, however, were essentially the same as diseased cases. Given the large number of cases in this study and the wide variety of tissues and preparations analyzed, these findings demonstrate the utility of the PMS as a tool for evaluating the metabolic status of a tissue or organ. Note: there are cases where tissue is diseased but where the PSM is essentially unchanged from healthy tissue because the overall energy metabolism of the tissue is not affected by the disease. Alternatively, there are cases where diseased dystrophic tissue is metabolically enhanced (Figure 4).

In order to measure the fidelity of the measurements made by the digitization of the UN-SCAN-IT graph used to compute the PSM, the integral of the spectral run obtained from the NMR spectrometer and in cases where the actual numbers computed in mole percent phosphorus or gm/mmole phosphate were reported, showed no statistical differences among these values (Table 2).

## 4. Discussion

### 4.1. Selection of ^31^P Spectra for PSM Computation

The inclusion criterion required using only those articles presenting full ^31^P spectra from 10 δ to −25 δ; therefore, any study articles that focused on ^31^P NMR spectra of individual molecules or molecular groups were not incorporated into this meta-analysis.

### 4.2. Reference Markers in ^31^P NMR Spectra

Where ^31^P NMR spectra were presented that included standard reference markers e.g., phosphonates, such as 0.2 M methylenediphosphonate [120] used to calculate metabolite concentrations and that come into resonance at a region of the spectrum >15 δ, the reference region of the spectrum was not included in the calculation of the PSM. The calculation of the PSM only included that area of the ^31^P spectrum lying between 10 δ to −25 δ.

In studies where ^31^P NMR spectra included reference markers within the spectral area of the sample spectrum (10 δ to −25 δ), e.g., inorganic orthophosphate (Pi) contained in supporting media and, for example, used to determine sample tissue pH, such spectra were not used because the signal intensity of such reference compounds could not be precisely determined.

### 4.3. ^31^P NMR Spectra Computational Considerations

Using published ^31^P NMR spectra in order to calculate the PSM, the following parameters may present problems in accomplishing computation. Calculation of the PSM is best when a scale depicting the resonance shift positions is present. Spectra presented with an inaccurately positioned horizontally displayed scale or that may be presented where the scale is shifted unreliably or the signs are reversed, can potentially result in misidentification of a phosphatic metabolite. In most instances, using appropriate corrections, such spectra can be precisely integrated for determination of the PSM. Among spectra derived from published manuscripts, measurement may have interrupted or discontinuous line tracings depicting spectral signals that may require completion prior to UN-SCAN-IT analyses. High-resolution signals with high signal-to-noise may preclude use of the computation program due to the thinness of the spectral signal lines precluding measurement; signal lines also may be excessively thick, prohibiting computational measurements; signal heights may be truncated. Spectra may be poorly resolved, such that no clear inflection point can be determined between the low-energy and high-energy phosphate bands. Baseline spectral noise segments lying on either side of the region of interest may be too short to allow determination of a reasonable base line segment (less than 5% of the entire spectrum), or presented as nonexistent, or the spectral tails were elevated and distorted, or the spectra were markedly elevated in the central portion of the spectrum, or there was presence of shading, cross-hatching, or backfilling of the signal areas, or the background on which the spectrum was presented had a granular texture, or was shaded or gray in its entirety interfering with the UN-SCAN-It digitizer computational program, or vertical demarcation lines or diagonally oriented lines passing through either the upfield or downfield arms of a spectral signal [5], or any other descriptors or labeling that were included in the spectral figure and that interfered with the digitization process. In these cases, additional lines, descriptive designations, or labels require that they be obliterated prior to UN-SCAN-IT analyses. This of course is not necessary where spectra are created and subjected to UN-SCAN-IT analyses prior to adornment of the spectra with such features.

### 4.4. Low Signal-to-Noise, Signal Resolution or Number of Reported Phosphatic Metabolites

The gathering of ^31^P NMR reference data was not restrictive and allowed for calculation of the PSM even when based on the presence of a low number of reported phosphatic metabolites (low signal-to-noise) or a low level of ^31^P spectral resolution (separation of the two component bands). This was allowed so that data from reports with low signal-to-noise or a signal exhibiting poor signal resolution might be included. Reference phosphates added to samples that had already been scanned allowed precise biochemical identification of specific resonance signals, where: data was calculated from cells, tissues, and organs under various conditions/stressors (Table 2 and Appendix A); the effects of conditions/stressors, such as lens aging [121], hypoglycemia [13], and hyperglycemia [19], were clearly present, insufficient tissue perfusion was stated or apparent, were manifest.

Since the anhydride phosphate bonds (e.g., the high-energy triphosphate chain of ATP) are necessary to conduct metabolic functional requirements as well as mediate the prevention of protein aggregation of cells, tissues, and organs, the maintenance of an elevated PSM is important, as it is an assessment and assay of metabolic status. In general, when the PSM is maintained above a numerical value of >1.3, it is indicative of an organism’s cell’s, tissue’s or organ’s capacity to maintain a balance of adequate high vs. low metabolic health. In diseased tissue, or where the PSM is reduced significantly, there may not be sufficient high-energy phosphate to support good metabolic health, and cell/tissue/organ function will begin to be compromised. As such, the metabolic health status measured by the PSM assay becomes very useful for determining tissue metabolic health and disease especially when considering that in disease, metabolic abnormalities will usually or typically predate observable signs of anatomical pathology.

### 4.5. Metabolic Changes

Metabolic changes may be detected prior to any changes detectable by physical or clinical examination. An example of this includes a healthy transparent crystalline lens versus a cataractous lens with physically observable opacification detected in vivo clinically by slit lamp biomicroscopic or ex vivo by photomacrography. The lens may maintain its function by being transparent but may have mild or even severe metabolic damage as detected by ^31^P NMR in advance of any detectable physical sign of loss of transparency [8,122]. In disease states, a decline in the PSM can be observed before the detection of changes observed by physical examination or biomicroscopically. The PSM also can be a measure or predictor of disease prognosis, as evidenced by the reduction in the PSM value monitored over a time-course.

### 4.6. ^1^H and ^31^P NMR

Although the PSM can be calculated when the concentrations/quantifications of a total ^31^P spectrum of organophosphates can be easily measured, in cases where it is difficult to detect and measure distinct organophosphates or anything but an integral curve, for example where a low signal-to-noise ratio is obtained, the integral curve will suffice. Thus, the PSM, computed using the NMR generated integral curve, has the advantage of requiring less instrument time for the measurement The detection of phosphate (^31^P) signals from a lower signal intensity in situ contrasts with proton (^1^H) NMR, which produces a far greater signal-to-noise. This difference does not arise from the abundance of ^31^P nuclides but from the concentration of phosphates in the tissues. (^31^P is 100% naturally abundant as, essentially, is ^1^H). The responsible factor for this difference sensitivity is the magnetogyric ratio, which is much larger for hydrogen (^1^H, 42.58 MHz/T; ^31^P, 17.24 MHz/T), thus generating a proportionately greater signal for the same number of nuclei. Moreover, the concentration differences of ^1^H nuclei versus ^31^P nuclei in a cell, tissue or organ is the major reason why ^31^P is so much less useful than ^1^H for imaging tissues. These are the important factors when examining a tissue with an NMR surface coil tuned to detect ^31^P with its limitation of signal strength as well as that of resolution, which may only allow quantification through detection of an integral curve. However, with detection of a ^31^P NMR spectral integral curve using surface coil technology, calculation of the PSM is possible.

Finally, it is fortunate that high-resolution ^31^P NMR measures only the low molecular weight metabolites in cells, tissues, and organs, and not the nucleic acids, phosphorylated proteins, or the phospholipids contained in membranes [1]. This is because the ^31^P relaxation times in molecules such as DNA and RNA and phospholipids in membranes are short, such that under high-resolution scan conditions, their signals are extensively broadened, becoming part of the baseline for the high-resolution signals that are detected.

### 4.7. In Vivo Metabolic Assessment

Since determination of the integral curve permits calculation of the PSM for in vivo metabolic analysis of diseased cells, tissues, and organs, using this index, changes occurring during a pathogenic process may be monitored over time (Figure 5). This provides a biochemical assessment of tissue change at the metabolic level, as, for example, when studying pharmaceutical, dietary, elemental, nutrient, environmental, thermal, and aging effects on cells, tissues, and organs. When incorporated into magnetic resonance imaging (MRI) technology, this biochemical assessment may be applied to specific anatomical regions or specific tissues, thereby augmenting the value of the present ^1^H MRI procedure.

In vivo metabolic assessment of tissues and organs can be accomplished using the higher magnetic fields of the modern MRI instrumentation used clinically. Localization of specific tissues and organs can be determined using the conventional proton (^1^H) MRI image followed immediately (or simultaneously) with metabolic assessment using ^31^P NMR spectroscopy. Currently in developmental laboratories, these determinations can be accomplished using a high-field MRI equipped with a surface-coil dual tuned to ^31^P and ^1^H signal detection. The ^31^P MRI component used with the surface coil will gather signals detected, such that each scan may be integrated to provide relative quantification of the spectral low-energy and high-energy phosphate bands. Computation of the PSM from each acquired ^31^P MRI spectrum so integrated may then be used to compute a pixel array of the PSM values for the creation of a ^31^P image of the tissue under assessment.

It has been established from our static ex vivo time-course studies of the crystalline lens phosphorus chemistry that such information is useful in assessing changes in biochemistry of intact tissues (Figure 3) [18]. Differences in PSM values (Table 2) allow for the ability to differentiate normal vs. disease or stressed tissues in situ. Such differences have already been demonstrated with the crystalline lens in situ, where the decline in the PSM value correlated directly with the degree of loss of visual acuity due to decreased lens transparency [124]. The PSM may be used to study methods of delivery of high-energy phosphates or increased production and supplementation of ATP. Similarly, declines in ATP overtime can be measured as effects of metabolic stressors and the loss of vision or transparency of the lens [13,18,19,122]. The use of the PSM offers a great advantage in the measurement of tissues in situ and in development of pharmaceuticals and other treatments on health and disease. We anticipate that useful data can be obtained using such analysis to study pharmaceutical, dietary, elemental, nutrient, environmental, and aging effects on cells, tissues, and organs. Using the eye as a model, we employed use of the PSM to first present the moduli in normal [4,123], stressed, or diseased tissues [123]. The PSM also can be used in the study of tissues harvested, grown, and then banked for eventual transplantation.

## 5. Study Strengths

There are three major reasons for using the PSM measurement: (1) It does not require high resolution magnetic spectral analysis, as the integral curve alone can be used to measure cells or tissue metabolic status. As such, a strong signal-to-noise ratio is not necessary to calculate the PSM. (2) The integral curve measured by ^31^P NMR can be used to detect metabolic damage in advance of detectable changes in normal, diseased, or stressed cells, tissues, and organs as would be found anatomically by physical examination, biopsy, and imaging or as detected biochemically by examination of body fluids with measurement of biomarkers. Use of the PSM provides the opportunity to use current surface-coil technology and dual tuned ^31^P and ^1^H NMR receiver systems to monitor metabolism without the need to quantify individual ^31^P signal resonances. (3) The use of nuclear magnetic resonance can be repeated multiple times during the time-course of a normal tissue that is being sustained, normal tissues subjected to stress, and diseased tissues without cellular or tissue injury. Repetitive NMR measurements utilizing electromagnetic waves are un-harmful, since there is no requirement of roentgen radiation accompanying other computerized tomography technologies.

### 5.1. Overcoming Signal-to-Noise Ratios

Proton NMR technology or NMR imaging (MRI) utilize measurements of the element hydrogen to create an image. However, there are underlying biochemical changes in hydrogen prior to detection of pathological damage which can be measured at the metabolic level using ^31^P NMR. Unlike detection of changes at the level of tissue anatomy using ^1^H NMR, the use of ^31^P NMR is not able to allow detection of useful diagnostic images. This is because the concentration of body phosphorus is but a fraction of the high concentration of body hydrogen. Therefore, imaging with phosphorus is unable to resolve anatomy in detail. However, with the lower concentration of body phosphorus relative to the element hydrogen using ^31^P NMR, metabolic damage can be detected. As such, the ^31^P NMR scan time is extended in order to accumulate enough data points in order to make measurements at the metabolic level. However, this problem can be overcome multiple ways: (1) increasing the strength of the magnetic field, (2) extending data scan time and number of data point acquisitions, (3) altering the detection receiver with the use of a surface coil, and, (4) as described herein, utilizing the ^31^P spectral modulus derived from an integral curve instead of relying on accumulating data from individual organophosphate signals.

The ability to make measurements with the PSM is a fundamental advance in monitoring the efficacy of treatment of diseases. PSM technology can be applied to animals, plants, cells, and bacterial microorganisms including cell cultures. This technology allows seemingly endless possibilities for the noninvasive, nondestructive, non-toxic, and early detection of changes in the metabolic status of stressed or diseased cells, tissues, and organs.

### 5.2. The Advantage of Measuring Uncontrolled Input Data

In our previous report on the ^31^P spectral modulus, we restricted our observations to work performed in our laboratories where sampling, preparation, and instrumental methods were rigorously controlled [17], the analyses herein described were generated with totally uncontrolled input data published in the literature and still demonstrated reasonable statistical difference and power among normal and diseased, and stressed cells, tissues, and organs. This power of statistical treatment is expected to be much higher when analyses are carried out in a controlled setting, e.g., a hospital or medical center where NMR facilities equipped with surface coils calibrated to ^31^P, exist. Controlled conditions are expected to add precision to the PSM measurement. With the accumulation of patient data measurement there will be greater precision to the PSM measurement and the variance in the precision should become considerably smaller approaching an asymptote. As the asymptote is approached, there will be improvement in accuracy. As precision of the PSM measurement increases, measured values approach the actual value that reinforces and strengthens the procedure for PSM measurements.

## 6. Study Limitations

Our method for determining the PSM calculations included the following limitations: (1) Spectra were excluded where interfering external chemical reference markers were used to calibrate phosphorus chemical-shift positions on the delta scale (δ). This is because accurate calculation of the PSM would require subtracting the concentration of the marker. [Such markers are chosen because their chemical characteristics are affected less by physical chemical parameters within the sample specimen; a common marker, glycerylphosphorylcholine (GPC), is such a reference standard (−0.13 δ)] [3]. As such, the present paper includes only those ^31^P NMR studies where full-spectrum data sets without interfering markers were used. (2) Previously published spectra that include gray scale shading or pattern designs of individual or groups of individual spectral signals or creation of spectral signal symbols or borders were used to denote or surround a signal or a group of signals. (3) Spectra showing truncation of signal amplitude in order to accommodate limiting journal space requirements. (4) Spectra were presented in multiple layers-of-signal groups. (5) Spectra blocked together in a fashion so as to depict a series of spectra. (6) Full spectra were missing or did not include both low-field (10 δ to −0.13 δ) and high-field (−0.13 δ−25 δ) spectral segments. (7) Excessive signal-to-noise ratios interfered with integral curve digitation. (8) Included signal labeling of any kind interfered with quantification of the moduli and/or required that labeling be removed prior to integral curve digitizing software program calculations.

Considering other moduli, there is no other known modulus that is as encompassing for the measurement of cell or tissue metabolism than the PSM. This PSM is of increasing importance with the ever-increasing ability of magnetic resonance technology to measure the metabolic health status of targeted tissues in vivo. This is a direct result of increasing magnet field strength and programing allowing higher signal-to-noise determinations. Increasing magnet field strength and improved signal-to-noise allows an ever-increasing ability to measure the metabolic health status of living tissue with ^31^P magnetic resonance technology.

## 7. Future Studies

With increases in the number of cases in the PSM database, the higher will be the fidelity of the PSM number, with a consequential increase in diagnostic precision. The PSM is not a qualitative assessment like a radiographic, ultrasonographic, or proton NMR image, which requires training and visual expertise for interpretation. This PSM is a statistical evaluation. It is a living index of tissue health.

## 8. Summary

In summary, the importance of understanding the usefulness of the PSM is increasing and can presently be performed utilizing the integral curve analysis. When analysis of ^31^P NMR spectra involves only computation of the PSM, it is not necessary to employ ^1^H decoupling. ^31^P-^1^H couplings in biological phosphates are small, on the order of 5 Hz, and even with proton coupling present, there is more than enough separation between the high-energy phosphate and low-energy phosphate spectral bands for integration of these two spectral signal groups. The separation of spectral bands permits calculation of the PSM even with proton coupling. The PSM serves as an assay of metabolic health with the healthy cells, tissues, and organs having a high PSM and the diseased or stressed tissues having a lower or declining PSM.

The importance of the use of the PSM in experimental studies and clinical and basic science studies cannot be overlooked or underestimated. The utility of the phosphorus-31 spectral modulus (PSM) provides a noninvasive and nondestructive capability of assessing metabolic health status and monitoring clinical and diagnostic medicine during a time-course measuring the efficacy of therapeutic treatment regimens at the level of cellular, tissue, and organ metabolism.

The PSM is not a qualitative assessment, it is a statistical evaluation and can serve as a metabolic index of cell, tissue, and organ health, disease, and stress, in vivo, ex vivo, and in vitro. The PSM represents a new parameter in medicine’s quest for encompassing in vivo, ex vivo, and in vitro measurement of the metabolic status in cells, tissues, and organs. Moreover, the PSM measurement offers a fundamental change in the ability to monitor the efficacy of the treatment of diseases, monitoring normalcy, and the effects of stressors determining the metabolic status of cell, tissue, or organ systems.

## Figures and Tables

**Figure 1 biology-14-00152-f001:**
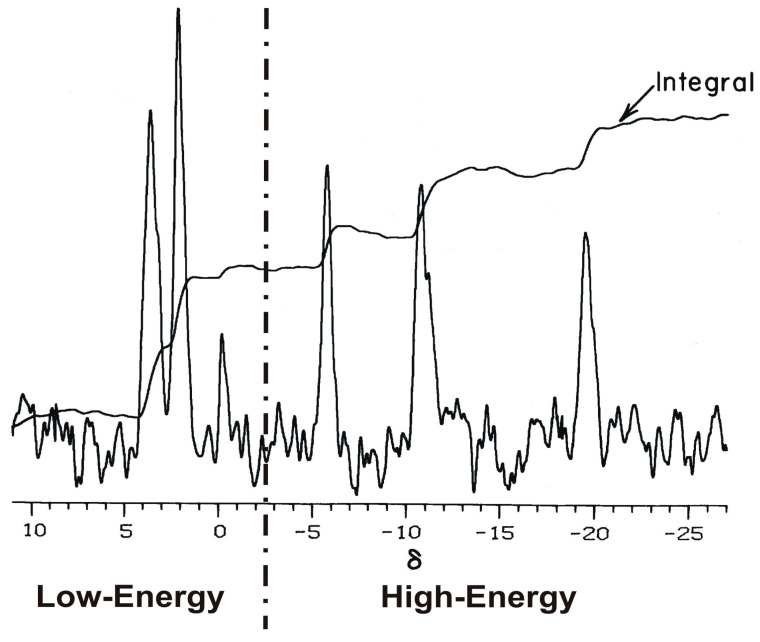
Phosphorus-31 nuclear magnetic resonance (^31^P NMR) spectrum of a living human cornea observed with 20 Hz spectral filtering [4]. The spectral integral (arrow) also is presented overlying the NMR spectrum. The chemical shift scale, δ, is presented relative to 85% orthophosphoric acid as recommended by the International Union for Pure and Applied Chemistry [6]. The vertical line divides the spectrum and integral into low-energy phosphates and high-energy phosphates. The low-energy phosphate band is composed principally of resonance signals of sugar orthophosphate esters. The prominent three signals in the high-energy phosphate band arise principally from the α, β, and γ phosphates of adenosine triphosphate (ATP), respectively, at −10.8, −19.5, and −5.8 δ. Of the total phosphate resonance signals detected, ATP provides 42% of the detected signal, which is a quantity similar to that ordinarily observed with other healthy living tissues [7]. Most, but not all, biological high-energy phosphates contain phosphoric anhydride functional groups, which mainly account for their chemical shift positions.

**Figure 2 biology-14-00152-f002:**
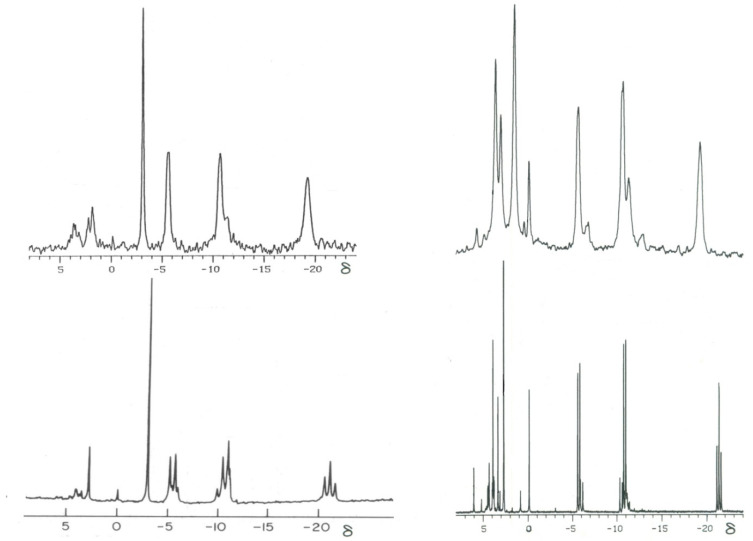
Phosphorus nuclear magnetic (^31^P NMR) spectra obtained from a dog heart (**left**) and pig lens (**right**) [3]. Top spectra were obtained, respectively, from the freshly excised intact organs, and bottom spectra from perchloric acid (PCA) extracts of these same organs. The ^31^P high- and low-energy spectral bands are grouped, respectively, into corresponding chemical shift (δ) ranges between −0.13 δ to −25 δ and between 10 to −0.13 δ, demonstrating concordance, both qualitatively and quantitatively, between each organ’s live tissue spectrum and its PCA extract spectrum. The identity of the signals can be garnered by comparing the position of the signals on the chemical shift scale (δ) with the chemical shifts in Table 1. The single tall signal in the dog heart spectra arises from phosphocreatine (PCR); the three prominent signals to the right of this signal arise, principally, from the γ, α, and β adenosine triphosphate (ATP) phosphates in the chemical shift range of, respectively, −5, −10, and −20 δ. The pig lens spectra exhibit similar spectral profiles, with the exception that the sharp signal at 0 δ arises from phosphatidylglycerol and only a trace of PCR is detected in the PCA spectrum at −3.1 δ.

**Figure 3 biology-14-00152-f003:**
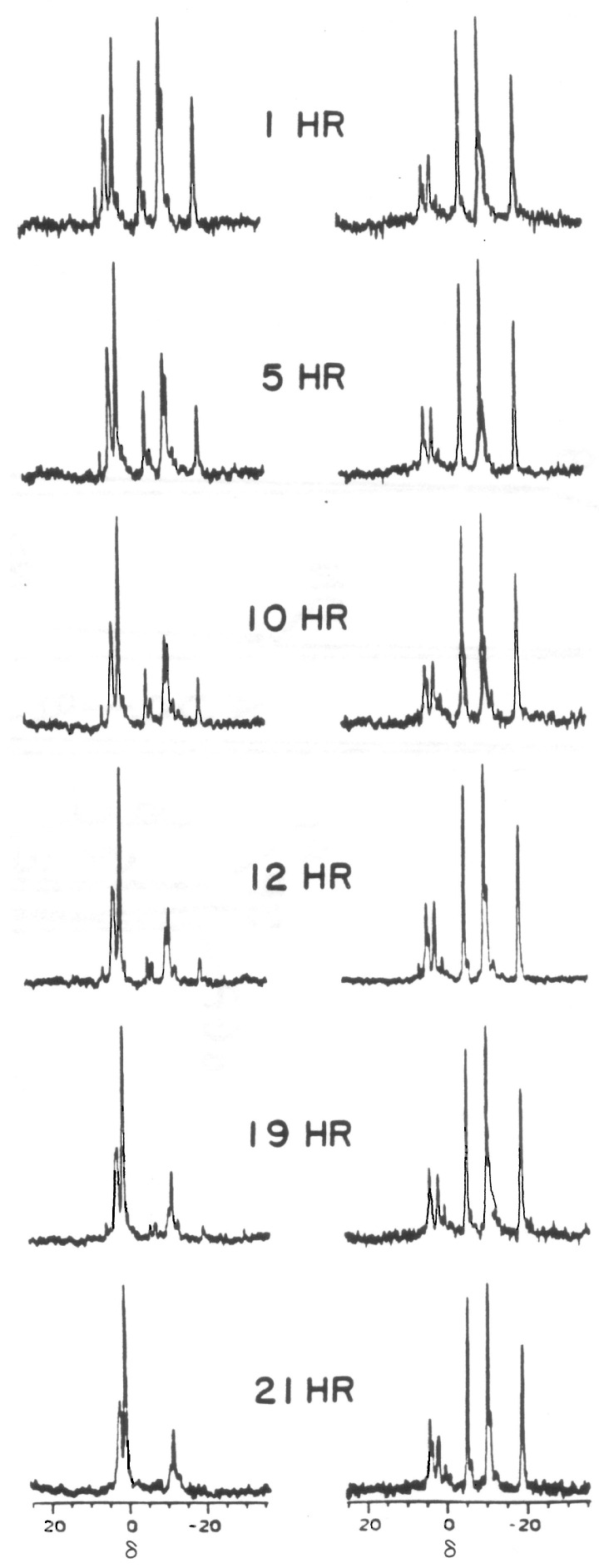
Phosphorus nuclear magnetic resonance (^31^P NMR) spectra of an ex vivo rabbit lens (intact organ) acquired at six periods during a time-course in hours (HR) involving treatment (left spectral column) of the lens with 2 × 10^−3^ M dexamethasone steroid, which is toxic to the lens. For comparison, the time-course of a corresponding control lens sustained with the same modified Earle’s buffer, but without added steroid, is presented in the right spectral column [18]. With time, the adenosine triphosphate (ATP) signals in the high-energy band of the steroid-treated lens spectra diminish and eventually disappear along with the other high-energy phosphates, while the phosphates of the low-energy band correspondingly increase whereas the control lens is stable.

**Figure 4 biology-14-00152-f004:**
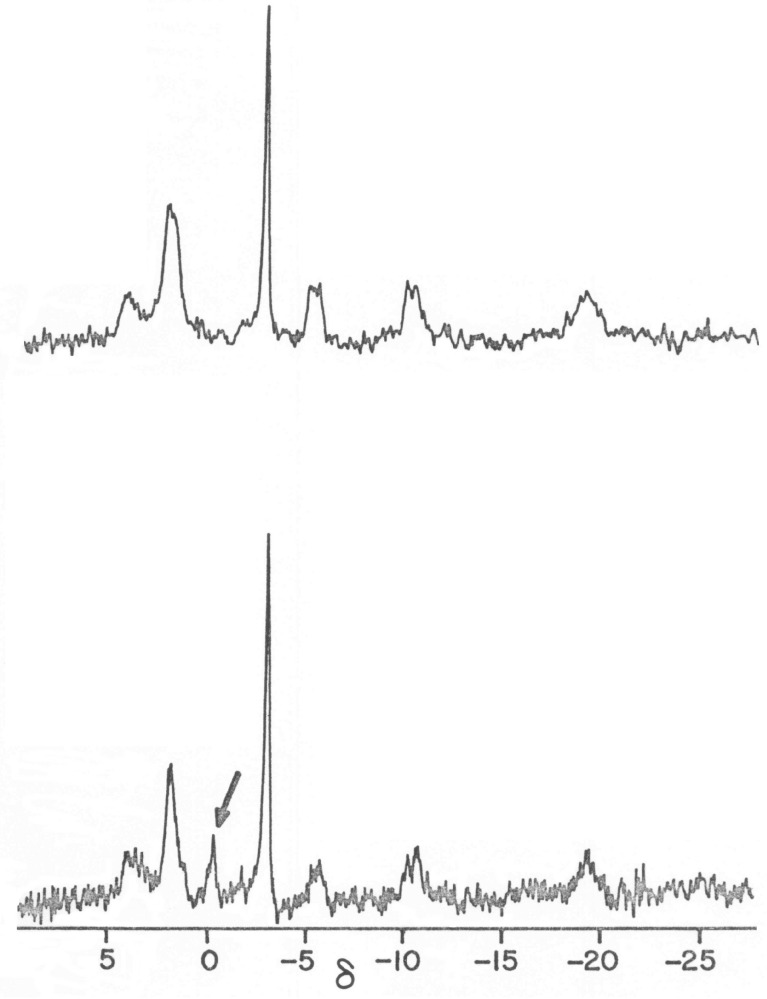
Comparison of phosphorus nuclear magnetic resonance (^31^P NMR) spectra of ex vivo normal (PSM: 1.022) and dystrophic chicken pectoralis (PSM: 1.406) muscles [20]. The arrow ((**bottom**) spectrum) points to a phosphodiester resonance signal in the dystrophic chicken muscle, the principal component of which is serine ethanolamine phosphodiester, which is present for all examples of such diseased muscles including human muscular dystrophy. This resonance does not appear in the corresponding healthy chicken pectoralis muscle (**top**). The other phosphate signals of both tissues, however, are essentially identical in appearance, indicating that the disease does not affect high-energy metabolism.

**Figure 5 biology-14-00152-f005:**
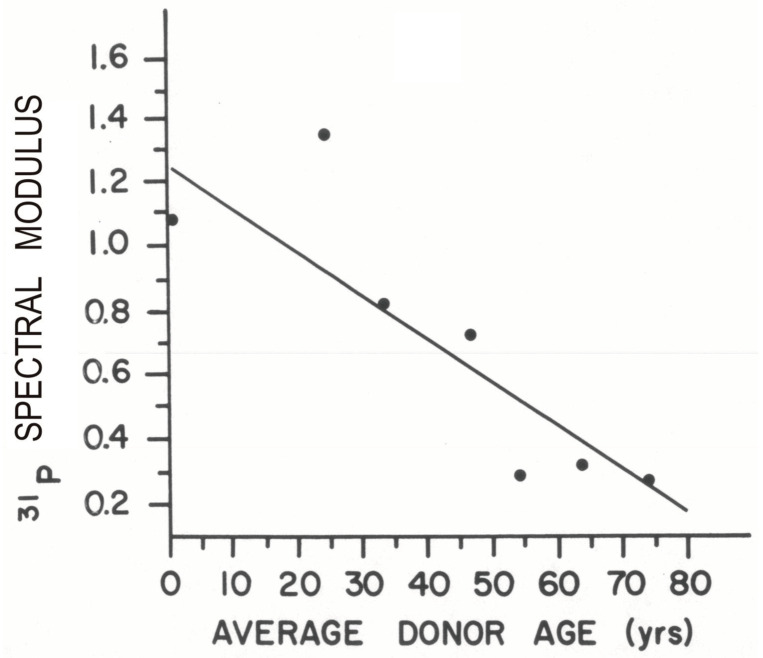
Human cornea phosphorus spectral modulus (PSM) as a function of donor age. With time, the human cornea’s high-energy phosphates diminish, with consequential diminution of the PSM. Each bullet represents an average (yrs) for each decade with a minimum of 10 corneas per perchloric acid extract [123]. The ^31^P spectral modulus correlated highly with age with a coefficient of determination of 79%.

**Table 1 biology-14-00152-t001:** ^31^P NMR chemical shifts of biological phosphates in solutions at pH 10 at 24 °C *.

Low-Energy Phosphates (Below)			
Compound	Chemical Shift (δ) ^†^	Compound	Chemical Shift (δ) ^†^
Glucose 6-P	4.44	Galactose 1-P	2.45
Mannose 6-P	4.32	Glucose 1-P	2.32
Glyceraldehyde 3-P	4.30	N-Acetylglucosamine 1-P	2.04
Glycerol 3-P	4.29	Glycerylphosphorylethanolamine	0.95
Fructose 1-P	4.29	Glycerylphosphorylglycerol	0.92
Dihydoxyactone P	4.16	Glycerylphosphoryl(monomethyl) ethanolamine	0.86
Galactose 6-P	4.14	Serine-ethanolaminephosphodiester	0.83
3(−) Phosphoglyceric acid	4.07	Glycerylphosphoryl(dimethyl) ethanolamine	0.80
Fructose 1,6-diP	4.05	Di(glycerylphosphoryl)glycerol	0.79
	3.91	Glycerylphosphorylserine	0.69
Fructose 2,6-diP	3.99	Glycerylphosphorylinositol	−0.07
	−0.43	Glycerylphosphorylcholine	−0.13
3′-AMP	3.96	Phosphoenolpyruvate	−0.68
Glycerol 2-P	3.92	Carbamyl P	−1.74
Phosphoserine	3.88	Acetyl P	−2.13
Ribose 5-P	3.85	**High-Energy Phosphates** **(Below)**	
Phosphorylethanolamine	3.84		
Fructose 6-P	3.83	Phosphoceatine	−3.12
5′-IMP	3.79	Phosphoarginine	−3.58
5′-XMP	3.78	ATP: γ ^↕^	−5.80
5′-AMP	3.77	α	−10.92
2,3-Diphosphoglycerate	3.76	β	−21.45
	3.43	ADP: β	−6.11
5′-CMP	3.69	α	−10.61
D(+)2-Phosphoglyceric acid	3.61	Nicotinamide dinucleotides ^‡^	−11.37
NADP,2′-P	3.52	CDP-Ethanolamine ^§^	−11.15
Phosphoglycolic acid	3.45	UDP-Galactose ^§^	−12.83
2′-AMP	3.40	UDP-Glucose ^§^	−12.99
Phosphocholine	3.31	UDP-Mannose ^§^	−13.70
Pi	2.61		

* Phosphates were ca. 0.01 M in P; the internal reference was glycerylphosphorylcholine (GPC). ^†^ Chemical shift values are reported relative to 85% inorganic phosphoric acid, with positive values corresponding to downfield shifts. ^↕^ The γ-group of GTP lies at −5.5 δ. ^‡^ Approximate center of the dinucleotide multiplets. ^§^ Approximate resonance band center. Shift values denote the centers of the hexose end-group phosphate doublet [4].

**Table 2 biology-14-00152-t002:** Statistical PSM comparisons of normal, stressed, and diseased groups (Column F).

Physiological State	Counts (*n*)	Minimum	Maximum	Mean	Std. Dev.	Statistical Comparisons (*t*-Test)	
						Pair	Probability
1, normal	165	0.115081	4.976203	1.650992	0.901282	nl vs. st	1.32 × 10^−5^
2, stressed	132	0.039337	3.885587	1.225661	0.752183	nl vs. dx	4.27 × 10^−3^
3, diseased	50	0.271944	3.062433	1.286694	0.727556	st vs. dx	0.61796
Total: 2&3	182	0.039337	3.885587	1.242428	0.743999	nl vs. (st + dx)	6.73 × 10^−6^

## Data Availability

The authors indicate that the data contained herein are available to anyone who wishes to use them.

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
