# Peer review of "The 31P Spectral Modulus (PSM) as an Assay of Metabolic Status"

_biology, 2025, doi:10.3390/biology14020152_

Round 1

Reviewer 1 Report

Comments and Suggestions for Authors

This manuscript explores the PSM as a tool for profiling the metabolic state of cells, tissues, and organs. Although this topic is interesting, there are some issues that need to be clarificated in this work.

1. The article selection are based on the authors' discretion, with articles removed due to particular spectral characteristics.

2. Relying solely on published literature to obtain 31P NMR spectra would bring about heterogeneity because each experiment could be subject to different experimental conditions.

3.  Remove any content that is not directly relevant to the main research question in the introduction, resulting in a more concise and impactful introduction.

4. In the introduction, it would be ideal to outline the PSM in relation to other widely used metabolic measuring techniques and its advantages.

5. The literature search was carried out in February 2023. There is already a significant time period to have elapsed and this may result in relevant publications being missed. It is best to perform an updated search.

6. Include a flowchart to depict the literature screening process.

7. The suggestion of using PSHM for in vivo assessments is premature. A careful investigation of the factors such as signal penetration, spatial resolution, and the effects of physiological activities on 31P NMR signals is required before the transition to in vivo.

8. Supplement the experimental section with a particular explanation of the NMR spectrometer used, like the model it is, the strength of the MR, and other relevant parameters.

Author Response

Please refer to the attachment for details.

Point 1. Regarding the statement on article selection, we did not select studies based on spectral characteristics. We selected studies based on the way each author presented their spectra in their figures and whether or not the UN-SCAN-IT Graph Digitizer Software could digitize the spectrum based on reasons we described in this paper, section 4.3 31P NMR Spectral Computational Considerations on page 13, lines 424-452 (revised manuscript page 13 lines 522-551).

Point 2. We agree with the reviewer that relying on published literature brings about heterogeneity because each experiment is subject to different experimental conditions. The PSM calculation, nevertheless, demonstrates the robust nature of this assay of metabolic status. In an actual laboratory or hospital setting, the investigators familiar with the NMR technology in that location, of course, must calibrate their instrumentation with appropriate standards before using them for studies or medicine.

Point 3. We agree with the reviewer and have reviewed the Introduction and removed content not directly relevant to the main research question. We thank the reviewer as this has resulted in a more concise Introduction removing lines 83-93, and we joined the line 83 (revised line 87) and 93 (revised line 88) such that the sentence now reads “The metabolic status is a determiner of the health of cells, tissues, and organs and thus can be used to monitor therapeutic and non-therapeutic health via the integral curve. The quantitative signal areas obtained represent information at the cellular level arising from the gross composition of cellular membrane intermediates, phosphorylated intermediates of intermediary metabolism and intermediary polynucleotide biochemistry, as well as the vitamin nucleotide enzymatic co-factors of metabolic processes.” (revised lines 87-92).

Additionally, we removed lines 99-102 (revised lines 94-132) under the section entitled 1.1 31P Spectral Metabolites on page 2,3, which now reads “The phosphatic metabolites included within the spectral range incorporated into the PSM computation (from 10 to -25 on the δ scale) must contain a phosphate chemical functional group, either as an orthophosphate ester or as an esterified condensed (anhydride) phosphate, such as ADP and ATP chain phosphates (usually esterified on one end). The exceptions are molecules, such as phosphocreatine and functionally related molecules, e.g., phosphoarginine, which have chemical properties similar to those of the phosphate anhydrides. The 31P spectrum has very few biochemical exceptions, e.g., the phosphonates, principally derived from marine sources, and phosphine and other reduced forms as products of microbial activity under anaerobic conditions [6]. Such molecules have 31P chemical shifts well outside the range used in the PSM computation. As many as 43 organophosphates of intermediary metabolism have been detected and measured spectroscopically in a single assay using 31P NMR (Table 1) [3].

Along with the above changes we also, increased the conciseness of the Simple Summary by adding a “.” After the word “spectra” on line 20 and the words “whether determined in vitro or in vivo.” after the word “metabolites” in line 21 on page 1 (revised page 1 lines 21,22).

In the interest of increased conciseness of the Abstract, we also added the words “and could be calculated solely from the integral.” after the word “spectrum” in line 35 on page 1 (revised line 36).

Point 4. We agree with the reviewer that it would be ideal to outline the PSM in relation to other widely used metabolic measuring techniques, however, this would require an enormous review of literature and research techniques, and this request is far and beyond the scope of the present paper and would require extensive explanations for each reference cited in Table S1. This is because for example, there are quite a number of ways that even a single organophosphate, such as ATP, can be measured. Such an undertaking would necessitate examining all the work on ATP, much of which can be found in our previous publications cited. Regarding our work detailing NMR parameters and methods, we are, however, encumbered to avoid adding any further references of our work as we are subject to a limit of self-citation of our previous published work as it had exceeded the suggested self-citation reference limit.

Point 5. Regarding the literature search, we have updated the original search run on February 15, 2023  to now extending to December 2025, employing our institutional librarian Dr. Collins at the Howe Library of Ophthalmology, Massachusetts Eye & Ear Infirmary. This search provided 6 additional references that were reviewed.  After review, none of these references qualified for inclusion in this study, and, thus, only the total number of references reviewed was changed from 507 to 513. We changed the number of studies reviewed to “513” on lines 235 and 305 (revised manuscript lines 290 and 370). Also, we changed the date of the literature search on lines 231 and 239 to include December 2025 (revised lines 286 and 294). Since there was no change in the number of references used in Table S1, there were no changes in Table 2 content and, therefore, no change in the statistical computations presented in the text.

Point 6. Although a flow chart depicting the literature screening process might be included, there were only three steps in the screening process, that is, “a total of 513 articles were reviewed, which includes the 6 additional reports now retrieved from the period February 2023 to December 2024, and after title/abstract and full-text screening, 99 articles remained for inclusion in the study, all articles were screened according to a screening which is described in section 4.3 31P NMR Spectra Computational Considerations explained and exhaustively detailed section from lines 425 through 452 from pages 13,14 (revised lines 503-531 pages 13,14). Moreover, the screening process was also described in section 2.5 Spectral Figure Data on page 7 lines 305-310 (revised page 7 lines 370-375). As such, a flow chart was not considered necessary and was not included.   

Point 7. The suggestion of using PSM for in vivo assessments is based on investigation of factors such as signal penetration, spatial resolution, and the effects of physiological activities on 31P NMR signals, which is presented in great detail in reference 4 by Glonek and Kopp. The technology required for in vivo work has been detailed and exact methods of surface-coil studies are well established.

Point 8. Regarding the addition of spectroscopic conditions that we have described in detail in many of the publications of our work, to now describe spectroscopic conditions in detail in each reference entry, if this information is necessary, the reader could resort to the cited reference in each case presented in Table S1.  The amount of work involved in completing this request would require rewriting the manuscript to include spectroscopic conditions for each and every case. The spectral figures analyzed were generated from numerous different spectrometers and conditions. Furthermore, not all of the references describe the experimental conditions in the required detail. It is unlikely that such a request is necessary, since the present paper involved 347 specimens along with reference citations detailing the appropriate methods, sections and calculations and the appropriate reference and table or figure for each reference.

Reviewer 2 Report

Comments and Suggestions for Authors

Dear authors,

Thank you for the scientific work you have done.

Overall, the work is interesting and has scientific and practical value. I have no comments regarding serious errors, but I would like to highlight a few points.

1) Despite the fact that the Authors claim that this method can work with spectra with a low noise level, it is necessary to exercise some caution when processing the results (for example, in Fig. 1). Within the entire integration section, the baseline during the absence of a signal has a very strong "variability" - it begins to decrease as in the case of signal emission. This situation means that the choice of the method of mathematical processing will lead to different values ​​of the final integral intensity, which violates the principle of scientific experimental data - repeatability. In order to increase the reliability of the results, it is necessary to conduct a certain sample of independent measurements, determining the boundaries of the confidence interval.

2) Figure 2 is presented very poorly from the point of view of spectroscopy. It is necessary, if possible, to provide an interpretation of the obtained spectra, indicating in the figure which line was obtained from which complex. The same goes for other drawings.

3) In Figure 5 there is too much scatter of points, there is no certainty of a linear dependence. Rather, the fit function may be much more complex.

Author Response

Please refer to the attachment for details.

Point 1. As can be seen from the references cited herein, the integration of these NMR spectral integral curves has been established for years and the methods well respected in the NMR analytical literature. We agree with the reviewer and answer in the affirmative, when it comes to limitations of the NMR analysis. The NMR analysis has limitations which vary from laboratory to laboratory depending on how it was done. Anyone that uses the PSM calculation herein described, would of course have to calibrate their NMR instrumentation against known specimens. From our own experience, this calibration can result in about a 5-fold increase in the precision of measurement. Considering the above, we did not determine confidence intervals.

Point 2. In Figure 2, reference [3] has been provided which cites detailed spectroscopic information. We have added the statement “The identity of the signals can be garnered by comparing the position of the signals on the chemical shift scale (δ) with the chemical shifts in Table 1,” on revised page 9 lines 428-430. Further, these signals have been previously reported in published literature where such identification has been provided [3].

Point 3. This Figure 5 was taken from literature reference number [121]. This is simply a referenced example of the type of data one can expect to obtain.

Round 2

Reviewer 1 Report

Comments and Suggestions for Authors

The author has provided a satisfactory response to the posed questions and I recommend publication.

Reviewer 2 Report

Comments and Suggestions for Authors

I am quite satisfied with the Authors' answers. The manuscript may be subject to further publish processing.